# Cellular Rejection Post-Cardiac Transplantation: A 13-Year Single Unicentric Study

**DOI:** 10.3390/medicina61081317

**Published:** 2025-07-22

**Authors:** Gabriela Patrichi, Catalin-Bogdan Satala, Andrei Ionut Patrichi, Toader Septimiu Voidăzan, Alexandru-Nicușor Tomuț, Daniela Mihalache, Anca Ileana Sin

**Affiliations:** 1Molecular and Cell Biology Department, George Emil Palade University of Medicine, Pharmacy, Science, and Technology of Târgu Mureș, 540142 Targu Mures, Romania; 2Pathology Department, Clinical Emergency County Hospital Targu Mures, 540136 Targu Mures, Romania; 3Pathology Department, Faculty of Medicine and Pharmacy, Dunarea de Jos University of Galati, 800008 Galati, Romania; 4Clinical Emergency County Hospital Braila, Pathology Department, 810325 Braila, Romania; 5Pathology Department, George Emil Palade University of Medicine, Pharmacy, Science, and Technology of Târgu Mureș, 540142 Targu Mures, Romania; 6Epidemiology Department, George Emil Palade University of Medicine, Pharmacy, Science, and Technology of Târgu Mureș, 540142 Targu Mures, Romania; 7Faculty of Medicine, George Emil Palade University of Medicine, Pharmacy, Science, and Technology of Târgu Mureș, 540142 Targu Mures, Romania

**Keywords:** ACR, AMR, cardiac transplantation

## Abstract

*Background and Objectives*: Cardiac transplantation is currently the elective treatment choice in end-stage heart failure, and cellular rejection is a predictive factor for morbidity and mortality after surgery. We proposed an evaluation of the clinicopathologic factors involved in the mechanism of rejection. *Materials and Methods*: This study included 146 patients who underwent transplantation at the Institute of Cardiovascular Diseases and Transplantation in Targu Mures between 2010 and 2023, and we evaluated the function and structure of the myocardium after surgery by using endomyocardial biopsy. *Results*: Overall, 120 men and 26 women underwent transplantation, with an approximately equal proportion under and over 40 years old (48.6% and 51.4%). Evaluating the degree of acute cellular rejection according to the International Society for Heart and Lung Transplantation classification showed that most of the patients presented with acute cellular rejection (ACR) and antibody-mediated rejection (AMR) grade 0, and most cases of ACR and AMR were reported with mild changes (13% or 10.3% patients). Therefore, the most frequent histopathologic diagnoses were similar to lesions unrelated to rejection (45.2% of patients) and ischemia–reperfusion lesions (25.3% patients), respectively. *Conclusions*: Although 82.2% of the transplanted cases showed no rejection (ISHLT score 0), non-rejection-related lesion-like changes were present in 45.2% of cases, and because more of the non-rejection-related criteria could be detected, it may be necessary to adjust the grading of the rejection criteria. The histopathologic changes that characterize rejection are primarily represented by the mononuclear inflammatory infiltrate; in our study, inflammatory changes were mostly mild (71.9%), with myocyte involvement in all cases. These changes are associated with and contribute to the maintenance of the rejection phenomenon.

## 1. Introduction

Cardiac transplantation is currently the treatment of choice for patients with end-stage heart failure, with improved graft survival largely attributed to advancements in immunosuppressive therapy. Although several non-invasive methods have been developed for the detection of graft rejection, endomyocardial biopsy (EMB) remains the gold standard for confirming the presence and severity of rejection [1,2]. The degree of cellular rejection is assessed according to the International Society for Heart and Lung Transplantation (ISHLT) grading system. EMB is performed either routinely, as part of post-transplant surveillance protocols, or in response to a clinical suspicion of graft rejection.

Given the limited availability of donor hearts, it is essential to optimize and standardize post-transplant protocols and immunosuppressive regimens to minimize the risk of graft failure and the need for retransplantation. Furthermore, reducing the frequency of EMB—especially within the first year after transplantation, when the risk of rejection is highest—is a desirable goal, both to improve patient quality of life and to enhance the efficiency of healthcare resource utilization [1].

Heart failure remains a major public health concern due to its high rates of morbidity and mortality [2]. The success of cardiac transplantation depends not only on effective immunosuppressive therapy, but also on the accurate histopathologic evaluation of rejection. Therefore, a deeper understanding of the mechanisms and histological patterns of cellular rejection is crucial, as rejection constitutes a key predictor of post-transplant outcomes, particularly in the early postoperative period. Short-term survival has markedly improved in recent years owing to enhanced immunosuppressive protocols, improved donor selection, and optimized graft preservation and retrieval techniques [3,4].

There is a limited number of studies addressing the phenomenon of graft rejection, particularly in the field of cardiac transplantation. Most existing research has been conducted on animal models, with relatively few investigations involving human subjects. Therefore, further research in this area is highly desirable, as it would contribute not only to advancing current knowledge, but also to developing new strategies for the treatment and management of transplant recipients.

In this context, we aimed to focus on the phenomenon of cellular rejection as well as antibody-mediated rejection, investigating the immunohistochemical markers that contribute to the maintenance of these processes. Additionally, we examined certain histopathological changes that may occur following graft transplantation, such as the Quilty effect and Cytomegalovirus infection.

According to the American Heart Association (AHA), antibody-mediated rejection (AMR) is defined as a host T-lymphocyte-mediated response mounted against the allograft tissue, and the ISHLT grading formula for the pathologic diagnosis of antibody-mediated rejection (pAMR) is based on histologic and immunologic criteria, as follows: pAMR 0 (negative for histologic and immunologic criteria), pAMR1 (histologic or immunologic criteria—CD68, CD4-positive), pAMR2 (histologic and immunologic criteria), and pAMR3 (severity criteria consisting of interstitial hemorrhage, capillary fragmentation, mixed inflammatory infiltrate, pycnose, karyorrhexis, edema) [5].

Acute Cellular Rejection (ACR) is defined according to ISHLT 2016 as Grade 0 = no signs of rejection; Grade 1R, mild = interstitial/or perivascular infiltrate with up to one focus of myocyte damage; Grade 2R, moderate = two or more foci of infiltrates with associated myocyte damage; and Grade 3R, severe = diffuse infiltrate with multifocal myocyte damage, with or without edema, hemorrhage, or vasculitis [6].

The Quilty effect is defined as the presence of a nodular endocardial infiltrate of lymphocytes in an endomyocardial biopsy (EMB) and has been described exclusively in heart transplant recipients [7].

Therefore, the aim of this study is to describe and analyze the histopathologic features observed in endomyocardial biopsies performed after cardiac transplantation, with a particular focus on the patterns and grading of cellular rejection. By doing so, we seek to highlight the diagnostic relevance of these findings and underscore the potential clinical implications of histologic interpretation in the management of transplant recipients. A better understanding of these patterns may support more targeted biopsy strategies, optimize immunosuppressive therapy, and ultimately contribute to improved patient outcomes in the early and long-term post-transplant periods.

## 2. Materials and Methods

We proposed an evaluation of the predictive factors involved in the mechanisms of cellular and humoral rejection. Thus, clinicopathological factors with significance in cardiac transplantation pathology and their interactions with different immunohistochemical parameters were evaluated.

The processing of the cases was approved by the Ethical Committee of the Clinical County Emergency Hospital of Targu Mures, Romania.

The inclusion criteria were as follows: patients diagnosed in the middle and late post-transplant period with different degrees of ischemia and reperfusion lesions, and patients with acute cellular and/or humoral rejection. Exclusion criteria were tissues unsuitable for processing, improper sampling of endomyocardial biopsy, and patients who died shortly after surgery, during the immediate post-transplant period when hyperacute rejection typically occurs, without performing any EMB. Moreover, patients who received their transplant in our center but whose post-transplant surveillance, including endomyocardial biopsy monitoring, was conducted at another institution were also excluded from the study.

For all cases, the available slides with cellular and humoral rejection or without signs of rejection were reanalyzed, and we aimed to grade them according to the most up-to-date grading system proposed by the ISHLT.

Based on the inclusion and exclusion criteria outlined above, a total of 146 patients were included in the study from an initial cohort of 172 cardiac transplant recipients at the Institute of Cardiovascular Diseases and Transplantation in Târgu Mureș between 2010 and 2023. In these patients, we evaluated myocardial function and structure postoperatively through endomyocardial biopsy.

A total of 449 endomyocardial biopsies (EMBs) were performed for the monitoring of rejection in these patients. Both pediatric and adult transplants were included in the study; however, given the limited number of pediatric cases at our center, we established an age cutoff for analysis, categorizing patients into two groups: those under 40 years and those over 40 years of age.

At our transplant center, preoperative induction therapy and immunosuppressive treatment are administered in accordance with ISHLT guidelines, and endomyocardial biopsy is performed following the standard procedure under fluoroscopic guidance [8].

The tissue analysis was performed in the Pathology Department of the Targu Mures County Emergency Clinical Hospital. All the interventions that were part of the present study were conducted with the informed consent of the patients. For endomyocardial tissue analysis, both the usual Hematoxylin and Eosin staining was used as well as special Tricrom Masson or Van Gieson staining to evaluate the degree of fibrosis, and immunohistochemical reactions to CD4, CD8, CD68, Ig M, Ig G, and CD20 were used to evaluate humoral rejection and the Quilty effect. Clinical–pathologic factors with relevance to the evolution of cardiac rejection were evaluated, as well as the degree of cellular and humoral rejection of each patient and its association with each clinical–pathologic parameter, according to ISHLT classification.

Statistical analysis: Microsoft Excel was used for data organization and descriptive statistics. Continuous variables, such as age, were reported as mean ± standard deviation (SD) and range. Age was estimated as the midpoint of the recorded interval ranges. All included cases were statistically analyzed using the GraphPad Prism (8th edition) platform. Odds ratios (ORs) with 95% confidence intervals (CIs) and associated *p*-values were calculated to explore potential associations between clinical and histopathological parameters. For associations between the clinicopathological features and ACR, AMR, Quilty effect, and IHC markers, the chi-square and Fisher’s exact test were used. A *p*-value under 0.05 was considered statistically significant.

## 3. Results

### 3.1. Clinicopathological Parameters of the Included Cases 

In our study, most patients were men and were over 40 years of age, with a mean age of 56 years (±15.8) (Table 1).

Most of the patients presented with ACR and AMR grade 0 (indicating a stable clinical status or absence of major post-transplant complications), with most cases of ACR and AMR being reported with mild changes (13% or 10.3% patients). Thus, the frequent histopathologic diagnoses were similar to those of lesions unrelated to rejection (45.2% patients, Figure 1) and ischemia–reperfusion lesions (25.3% patients), respectively (Table 1, Figure 1).

Quilty’s effect was observed in 12.3% of the studied population (Table 1).

Regarding the myocardial changes associated with post-cardiac-transplantation, the presence of fibrosis phenomena was observed, in most cases at extreme degrees; about 50% of cases showed a mild degree of fibrosis (62 patients), followed by the category of severe fibrosis (36 patients). Vasculitis was reported in 39 patients in the studied population (26.7%), and in the category of endothelial damage to intramyocardial small vessels, endothelial dystrophic lesions were the most frequently encountered (47 patients, 32.2%) (Table 1).

All cases had some degree of myocyte involvement, and many rejection cases had associated myocyte necrotizing changes (48.6%, 71 patients). Inflammatory infiltrate was present in the vast majority of endomyocardial biopsies (91.1%); 71.9% of patients showed minimal inflammatory changes (Table 1).

Over time, most cases saw resolved acute rejection/went into remission (93.2%), as evidenced by the increased frequency of mild acute rejection cases compared with the other categories (19 patients) (Table 1).

In the study sample, six cases with changes suggestive of CMV infection were also identified (4.1%) (Table 1).

### 3.2. Histopathological Diagnosis

#### 3.2.1. Acute Cellular Rejection (ACR)

Acute cellular rejection (ACR) was not influenced by the parameters of age or gender (*p* = 0.522, *p* = 0.411, respectively). Univariate logistic regression analysis did not reveal statistically significant associations between environment or CMV infection and ACR presence (*p* > 0.05 in all cases). A diagnosis of dilatative cardiomyopathy was significantly associated with lower odds of ACR (OR = 0.41, *p* = 0.046, CI: 0.17–0.98) (Table 2).

#### 3.2.2. Antibody-Mediated Rejection (AMR)

The distribution of AMR across clinical and demographic variables is summarized in Table 3. A statistically significant association was observed between acute humoral rejection (AMR) and patient age, with the population under 40 years of age reporting the most cases of humoral rejection at 18.31% (13 patients), which, compared with the population over 40 years of age—which had a frequency of 5.33% (four patients)—is approximately three times more. Also, male patients recorded twice the number of changes in AMR (11 patients vs. 6 patients) (Table 3).

CMV infection was also significantly associated with AMR. Among CMV-positive patients, 50% presented with AMR versus only 10% in CMV-negative individuals (OR = 9.0, *p* = 0.021), although the absolute number of CMV-positive cases was low.

While female sex and rural environment appeared to be associated with higher rates of AMR (23.08% and 15%, respectively), these differences did not reach statistical significance (OR = 0.34, *p* = 0.083 for sex; OR = 2.15, *p* = 0.200 for environment). No significant differences in AMR rates were observed between patients with dilatative versus restrictive cardiomyopathy (OR = 0.79, *p* = 0.789). No significant associations were found with environment (OR = 2.15, 95% CI: 0.72–6.46; *p* = 0.200).

#### 3.2.3. Quilty Effect

The presence of the Quilty effect was analyzed in relation to both clinical and histopathological parameters (Table 4).

Most cases with CMV infection were also associated with the presence of the Quilty effect, as the role of inflammation in the occurrence of any infection is well known, with a statistically significant result (*p* < 0.001, OR = 0.08, CI: 0.03–0.25), although the number of CMV-positive individuals was limited (Table 4).

There was also a statistically significant association between the presence of the Quilty effect and vasculitis changes, with twice the number of patients with the Quilty effect and associated vasculitis compared with those who presented only with Quilty changes without associated vasculitis (OR = 0.13, CI: 0.045–0.38, *p* < 0.001) (Table 4). Other features—such as inflammation (OR = 2.3, CI: 0.84–6.32, *p* = 0.158), myocyte damage (OR = 1.26, CI: 0.42–3.82, *p* = 0.769), and endothelial damage in intramyocardial small vessels (OR = 0.99, CI: 0.35–2.82, *p* = 1.000)—did not show statistically significant associations with the presence of Quilty lesions.

A statistically significant association was found between the type of cardiomyopathy and the Quilty effect. Patients with restrictive cardiomyopathy exhibited a substantially higher prevalence of Quilty lesions (42.31%) compared with those with dilatative cardiomyopathy (5.84%), with an odds ratio of 0.08 (*p* < 0.001, CI: 0.03–0.25).

While sex, age, and environment appeared to influence the presence of Quilty lesions, these associations did not reach statistical significance. Female patients had a higher prevalence (23.08%) compared with males (10%): OR = 0.37 (*p* = 0.094, CI: 0.12–1.10). Patients over 40 years more frequently had Quilty lesions than younger individuals (16.91% vs. 8%): OR = 2.34 (*p* = 0.132, CI: 0.83–6.62). An urban environment was also associated with a modest increase (15.16% vs. 10%), but the association was not statistically significant (OR = 0.62, *p* = 0.449, CI: 0.23–1.68).

### 3.3. Clinicopathological Features and Immunohistochemical Markers (Table 5)

An analysis of the association between clinic-pathological variables (sex and age) and immunohistochemical markers (Table 5) revealed several non-significant differences.

CD20 expression was found to be more frequent in female patients (23.07%) compared with males (10%), and was also slightly higher in younger patients (<40 years: 16.9%) compared with older ones (>40 years: 8%). However, these differences did not reach statistical significance (*p* = 0.094, CI: 0.12–1.10 and *p* = 0.132, CI: 0.83–6.62, respectively).

Similar patterns were observed for CD4 and CD8 markers, with both showing increased positivity in females and in patients under 40, but without statistically significant differences (*p* > 0.1).

CD68 expression showed a slightly higher rate in patients under 40 (28.16%) than in those over 40 (18.67%), with an OR of 1.71 (*p* = 0.240, CI: 0.79–3.72). For CD31 and CD34, their expression was also more common in younger individuals, but again non-significantly (*p* = 0.287, CI: 0.72–3.95 and *p* = 0.401, CI: 0.67–3.51, respectively).

For immunoglobulin markers (IgA, IgM), positivity was more frequent in younger patients and in females, though no associations reached statistical significance.

Lastly, EGFR showed a nearly equal distribution across sex and age groups, with no observable differences (*p* = 1.000, CI: 0.28–2.97 for sex, *p* = 0.815, CI: 0.47–3.00 for age).

Overall, none of the associations between immunohistochemical markers and patient sex or age were statistically significant in this cohort. However, certain trends suggest possible age- or sex-related variations that may warrant further investigation in larger studies.

## 4. Discussion

Currently, improvements related to diagnosis and treatment have made the one-year survival rate after cardiac transplantation about 90% in North America and 80% in Europe, with a median survival of more than 12 years [9].

However, knowledge of the existence of prognostic and predictive markers is imperative, given the few studies published on this topic, as the majority remain unknown.

Antibody-mediated rejection (AMR) of the graft heart transplantation is a rather challenging and incompletely elucidated diagnosis for both clinicians and pathologists, with very important prognostic and treatment significance. Broadly speaking, pAMR is characterized by the presence of known pathophysiological as well as immunopathological changes affecting complement activation, immunoglobulin binding, endothelial integrity, thrombotic environment (fibrin), complement inhibitors, and intravascular macrophages, which translate into positivity for the immunohistochemical markers Ig G, Ig M, CD3, CD4, CD55, CD59, CD31, CD34, and CD68.

CD4 is a marker used for the diagnosis of AMR; some authors consider it to be an immunopathologic surrogate for AMR [10,11,12,13,14,15,16,17,18]. It is most frequently associated with CD3; to better explain clinical graft dysfunction, this combination has been shown to be accurate in determining through immunofluorescence the prediction of graft dysfunction and mortality [19]. Also, in our present study, the two reactions were potentiated, with a positivity of 19.2% observed among the population studied; of this percentage, 17.1% were also associated with RAU.

Various studies have reported that the CD68 macrophage antigen allows for the detection of macrophage accumulation in vessels and differentiates them from lymphocytes, thus helping to exclude the existence of cellular rejection and, together with endothelial markers CD31 and CD34, establish the AMR diagnosis [10,20,21,22,23]. The presence of CD68-positive macrophages and the expression of endothelial markers CD31 and CD34 observed in the majority of AMR cases in our sample may reflect the vascular and inflammatory component typically associated with this type of rejection (Figure 2). Although our findings align with the existing literature, further studies on larger cohorts are needed to clarify the consistency and relevance of these immunohistochemical patterns.

The rejection phenomenon is the most common cause of post-heart transplantation graft damage, and ISHLT classifies this mechanism as one of two types: acute cell-mediated rejection (ACR) and antibody-mediated rejection (AMR).

Acute cell-mediated rejection (ACR) is broadly defined by the presence of mononuclear inflammatory infiltrate or myocyte damage, and in advanced stages by the appearance of vasculitis, edema, or interstitial hemorrhage (Figure 3) [24,25].

According to the revised ISHLT classification, the degree of acute cellular rejection is classified according to 0, 1R, 2R, or 3R. There are also lesions that may precede the cell rejection phenomenon, referred to as ischemia–reperfusion precursor lesions, which may occur shortly after surgery, and which also involve a complex mechanism coordinated by the interference of fibroblasts, leukocytes, endothelial cells, and cardiomyocytes [17,26]. In the present study, a rather high percentage of patients showed these changes (25.3%), with the first position being similar lesions not related to rejection (45.2%) (Table 1 and Table 2).

Immunosuppressive treatment is administered differentially, depending on the patient’s degree of ACR; stages 2–3 represent significant cellular rejection, which usually requires a change in immunosuppressive treatment or its adjustment [27,28]. A published study focusing on the evolution of the grade of rejection more than one year after cardiac transplantation revealed that the majority of patients (60%) were in grade 0 rejection, systematically followed by grade 1R (28.5%) [29]. In our long-term study, 82.2% of the patients were rejection-free (grade 0 ACR), while 19% had grade 1R CAR, with rejection categories 2 and 3 being at 4.1% and 0.7%, respectively (Table 1).

Although 82.2% of the transplanted cases showed no rejection (ISHLT score 0), non-rejection-related lesion-like changes were present in 45.2% of cases, and because more of the non-rejection-related criteria were detected, it may be necessary to adjust the grading rejection criteria.

The histopathological changes that characterize rejection are primarily represented by the mononuclear inflammatory infiltrate, which requires special attention. The study by Bu Weber et al. showed that lymphocytic inflammatory infiltrate may be present, even in a negative or early grade biopsy (1R), and can trigger the occurrence of a consecutive rejection or progression to an advanced grade of rejection [30]. In our study, inflammatory changes were mostly mild (71.9%), with myocyte involvement in all cases (Table 1). These changes are associated with and contribute to the maintenance of the rejection phenomenon.

This study presents a comprehensive analysis of potential associations between clinical, histological, and immunohistochemical parameters in the context of cardiac allograft pathology. While no statistically significant differences were observed in marker expression based on sex or age, some trends suggested lower odds of CD20, CD4, and CD8 positivity in males and a slightly increased expression of CD31, CD34, IgA, and IgM in patients under 40 years old. Among clinical predictors, restrictive cardiomyopathy and CMV positivity demonstrated increased odds for both AMR and Quilty lesions. Histopathologic findings such as vasculitis were strongly associated with the Quilty effect (OR = 0.13, CI: 0.045–0.38, *p* < 0.001). Overall, these results underscore the potential role of underlying diagnosis and histological injury in shaping graft immune responses, while suggesting that routine immunophenotypic markers are not substantially influenced by basic demographic variables in this context. Future studies should explore multivariable models and longitudinal trends to validate these insights.

## 5. Conclusions

Post-cardiac transplantation, routine follow-up of the patient is performed by means of endomyocardial biopsy, with the latter remaining an essential tool in the diagnosis of transplant pathology. It is imperative to evaluate all histopathologic parameters—in particular acute cellular and humoral rejection phenomena—as well as the associated changes in the rejection phenomenon, through both histologic, immunohistochemical, and molecular studies in order to initiate an appropriate immunosuppressive therapy. The present analysis provides a comprehensive evaluation of multiple clinical, immunohistochemical, and histopathologic parameters. Univariate logistic comparisons revealed no significant associations between sex or age and the expression of immunohistochemical markers (CD20, CD4, CD8, CD68, CD31, CD34, IgA, IgM, EGFR), although several markers showed directional trends. Notably, clinical risk factors such as CMV positivity and restrictive cardiomyopathy were associated with elevated odds of AMR and Quilty lesions, while histopathologic features such as vasculitis showed a strong correlation with the Quilty effect. These findings suggest that, while demographic variables may exert limited influence on marker expression, underlying clinical conditions and specific tissue injury patterns remain critical in the development of post-transplant immune alterations. The present study represents a starting point in this direction, but further studies are needed to understand and further investigate the mechanisms of cardiac graft rejection.

## Figures and Tables

**Figure 1 medicina-61-01317-f001:**
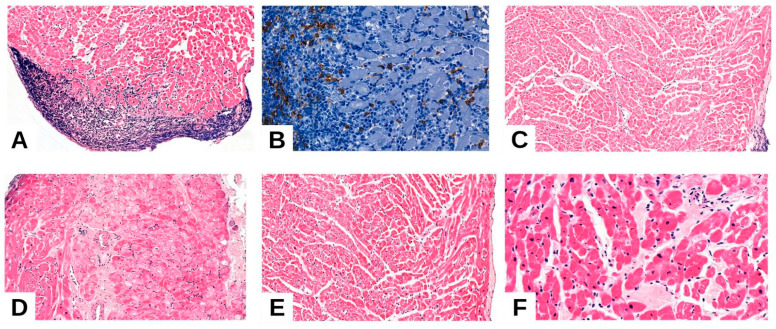
Quilty effect. (**A**). Quilty effect nodular aggregations of T, B-lymphocytes within the endocardium of the transplanted heart. (**B**) Quilty effect: nodular aggregates specifically positive to CD20 antibody. (**C**,**D**) ACR grade 1R and 2R—vasculitis: concentric mural thickening, endothelial dystrophic lesions (vascular endothelial swelling), inflammatory infiltrate, and perivascular edema. (**E**,**F**) Ischemia and reperfusion injury: Isolated subendocardial coagulation necrosis coexisting with edema and rare interstitial monocytic cells.

**Figure 2 medicina-61-01317-f002:**
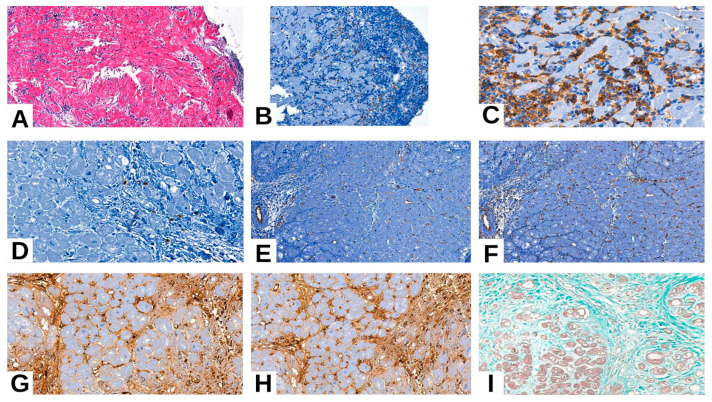
(**A**) ACR grade 1R: minimal interstitial lymphoplasmacytic inflammatory infiltrate. (**B**) ACR grade 1R: CD68-positive inflammatory chronic histiocytic infiltrate. (**C**,**D**) ACR grade 1R: CD4 and CD3 markers’ expression on T lymphocytes. (**E**,**F**) ACR grade 1R: no signs of vasculitis or AMR, with positive internal control of vascular endothelial markers CD31 and CD34. (**G**,**H**) ACR grade 1R with AMR features: inflammatory cell IgA and IgM expression. (**I**) ACR grade 1R: interstitial fibrosis stain with Tricrom.

**Figure 3 medicina-61-01317-f003:**
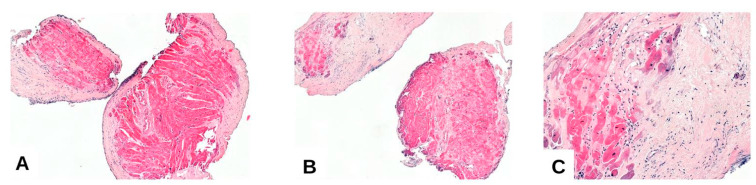
(**A**–**C**) ACR grade 3R: marked myocyte architectural disorganization: pronounced myocyte necrosis, myocyte degenerative lesions, diffuse interstitial inflammation, and perimyocyte extensive fibrosis.

**Table 1 medicina-61-01317-t001:** Clinicopathological parameters of the included cases.

	Parameter	*n* = 146
Mean age (years)	56 ± 15.8	
Median age	39 years	
Gender	Males	120 (82.2%)
	Females	26 (17.8%)
Age (years)	<40	71 (48.6)
	>40	75 (51.4%)
Quilty effect	Absent	128 (87.7%)
	Subendocardial	11 (7.5%)
	Endocardial	7 (4.8%)
Fibrosis	Absent	31 (21.2%)
	Mild	62 (42.5%)
	Moderate	17 (11.6%)
	Severe	36 (24.7%)
Histopathological diagnosis	ACR	26 (17.8%)
	Similar injuries unrelated to rejection	66 (45.2%)
	Ischemia and reperfusion injuries	37 (25.3%)
	AMR	17 (11.7%)
Vasculitis	Present	39 (26.7%)
	Absent	107 (73.3%)
Inflammation	Absent	13 (8.9%)
	Mild (1 focus)	105 (71.9%)
	Moderate (2 foci)	16 (11.0%)
	Severe (diffuse)	12 (8.2%)
Myocite damage	Degenerative lesions	35 (24.0%)
	Premiocitolisis	40 (27.4%)
	Myocite necrosis, coagulative necrosis	71 (48.6%)
Intramyocardial small vessels endothelial damage	Edema	49 (33.6%)
	Inflammation	21 (14.4%)
	Distrophic endothelial lesions	47 (32.2%)
	Fibrosis	29 (19.9%)
AMR	Histologic and immunopathologic studies are both absent (pAMR0)	129 (88.4%)
	Histologic findings or immunopathologic findings are present (pAMR1)	15 (10.3%)
	Histologic and immunopathologic findings are both present (pAMR2)	0 (0.0%)
	Severe pathologic AMR (pAMR3) (interstitial hemorrhage, capillary fragmentation, mixed inflammatory infiltrates, pyknosis, karyorrhexis, edema	2 (1.4%)
ACR	Absent (0)	120 (82.2%)
	Mild (1R)	19 (13.0%)
	Moderate (2R)	6 (4.1%)
	Severe (3R)	1 (0.7%)
Evolution	Resolved ACR	136 (93.2%)
	Remissive ACR	9 (6.2%)
	Continuous ACR	1 (0.6%)
CMV infection	Absent	140 (95.9%)
	Present	6 (4.1%)

**Table 2 medicina-61-01317-t002:** Clinical and Demographic Variables Associated with ACR.

Parameter		ACR Absent	ACR Present	OR	CI	*p*-Value
Sex	Male	83.33% (100)	16.67% (20)	0.67	0.24–1.87	0.411
	Female	76.92% (20)	23.08% (6)			
Age	<40	84.50% (60)	15.5% (11)	0.73	0.31–1.73	0.522
	>40	80% (60)	20% (15)			
Environment	Rural	81.25% (65)	18.75% (15)	1.15	0.49–2.72	0.830
	Urban	83.33% (55)	16.67% (11)			
Diagnosis	Dilatative Cardiomyopathy	87.09% (81)	12.91% (12)	0.41	0.17–0.98	0.046
	Restrictive Cardiomyopathy	73.58% (39)	26.42% (14)			
CMV infection	Male	66.67% (4)	33.33% (2)	2.42	0.42–13.95	0.290
	Female	82.85% (116)	17.15% (24)			

**Table 3 medicina-61-01317-t003:** Clinical and Demographic Variables Associated with AMR.

Parameter		AMR Absent	AMR Present	OR	CI	*p*-Value
Sex	Male	90.83% (109)	9.17% (11)	0.34	0.11–1.01	0.083
	Female	76.92% (20)	23.08% (6)			
Age	<40	81.69% (58)	18.31% (13)	3.98	1.23–12.86	0.019
	>40	94.67% (71)	5.33% (4)			
Environment	Rural	85% (68)	15% (12)	2.15	0.72–6.46	0.200
	Urban	92.42% (61)	7.58% (5)			
Diagnosis	Dilatative Cardiomyopathy	89.24% (83)	10.76% (10)	0.79	0.28–2.22	0.789
	Restrictive Cardiomyopathy	86.79% (46)	13.21% (7)			
CMV infection	Male	50% (3)	50% (3)	9.0	1.66–48.92	0.021
	Female	90% (126)	10% (14)			

**Table 4 medicina-61-01317-t004:** Clinico-pathological parameters and Quilty effect.

(**a**)						
**Parameter**		**Quilty Effect Absent**	**Quilty Effect Present**	**OR**	**CI**	** *p* ** **-Value**
Sex	Male	90% (108)	10% (12)	0.37	0.12–1.10	0.094
	Female	76.92% (20)	23.08% (6)			
Age	<40	83.09% (59)	16.91% (12)	2.34	0.83–6.62	0.132
	>40	92% (69)	8% (6)			
Environment	Rural	90% (72)	10% (8)	0.62	0.23–1.68	0.449
	Urban	84.84% (56)	15.16% (10)			
Diagnosis	Dilatative Cardiomyopathy	94.16% (113)	5.84% (7)	0.08	0.03–0.25	<0.001
	Restrictive Cardiomyopathy	57.69% (15)	42.31% (11)			
CMV infection	Male	16.67% (1)	83.33% (5)	0.83	0.09–7.51	<0.001
	Female	14.28% (20)	85.72% (120)			
(**b**)						
**Parameter**		**Quilty Effect Absent**	**Quilty Effect Present**	**OR**	**CI**	** *p* ** **-Value**
Vasculitis	Absent	94.39% (101)	5.61% (6)	0.13	0.045–0.38	<0.001
	Present	69.23% (27)	30.77% (12)			
Inflammation	Absent	80.48% (33)	19.52% (8)	2.3	0.84–6.32	0.158
	Present	90.47% (95)	9.53% (10)			
Myocyte damage	Absent	85.71% (30)	14.29% (5)	1.26	0.42–3.82	0.769
	Present	88.28% (98)	11.72% (13)			
Intramyocardial small vessel endothelial damage	Absent	87.75% (43)	12.25% (6)	0.99	0.35–2.82	1.000
	Present	87.62% (85)	12.38% (12)			

**Table 5 medicina-61-01317-t005:** Clinico-pathological features and immunohistochemical markers.

**Parameter**		**CD20 Positive**	**CD20 Negative**	**OR**	**CI**	** *p* ** **-Value**
Sex	Male	10% (12)	90% (108)	0.37	0.12–1.10	0.094
	Female	23.07% (6)	76.93% (20)			
Age	<40	16.90% (12)	83.1% (59)	2.34	0.83–6.62	0.132
	>40	8% (6)	92% (69)			
**Parameter**		**CD4 Positive**	**CD4 Negative**	**OR**	**CI**	** *p* ** **-Value**
Sex	Male	16.67% (20)	83.33% (100)	0.45	0.17–1.18	0.106
	Female	30.76% (8)	69.24% (18)			
Age	<40	22.53% (16)	77.47% (55)	1.53	0.67–3.51	0.401
	>40	16% (12)	84% (63)			
**Parameter**		**CD8 Positive**	**CD8 Negative**	**OR**	**CI**	** *p* ** **-Value**
Sex	Male	16.67% (20)	83.33% (100)	0.45	0.17–1.18	0.106
	Female	30.76% (8)	69.24% (18)			
Age	<40	22.53% (16)	77.47% (55)	1.53	0.67–3.51	0.401
	>40	16% (12)	84% (63)			
**Parameter**		**CD68 Positive**	**CD68 Negative**	**OR**	**CI**	** *p* ** **-Value**
Sex	Male	22.5% (27)	77.5% (93)	0.79	0.30–2.08	0.616
	Female	26.92% (7)	73.08% (19)			
Age	<40	28.16% (20)	71.84% (51)	1.71	0.79–3.72	0.240
	>40	18.67% (14)	81.33% (61)			
**Parameter**		**CD31 Positive**	**CD31 Negative**	**OR**	**CI**	** *p* ** **-Value**
Sex	Male	16.67% (20)	83.33% (100)	0.54	0.20–1.45	0.265
	Female	26.92% (7)	73.08% (19)			
Age	<40	22.53% (16)	77.47% (55)	1.69	0.72–3.95	0.287
	>40	14.67% (11)	85.33% (64)			
**Parameter**		**CD34 Positive**	**CD34 Negative**	**OR**	**CI**	** *p* ** **-Value**
Sex	Male	18.33% (22)	81.67% (98)	0.61	0.23–1.63	0.415
	Female	26.92% (7)	73.08% (19)			
Age	<40	22.54% (16)	77.46% (55)	1.53	0.67–3.51	0.401
	>40	16% (12)	84% (63)			
**Parameter**		**Ig A Positive**	**Ig A Negative**	**OR**	**CI**	** *p* ** **-Value**
Sex	Male	16.67% (20)	83.33% (100)	0.54	0.20–1.45	0.265
	Female	26.92% (7)	73.08% (19)			
Age	<40	21.12% (15)	78.88% (56)	1.41	0.61–3.27	0.523
	>40	16% (12)	84% (63)			
**Parameter**		**Ig M Positive**	**Ig M Negative**	**OR**	**CI**	** *p* ** **-Value**
Sex	Male	15.83% (19)	84.17% (101)	0.51	0.19–1.38	0.255
	Female	26.92% (7)	73.08% (19)			
Age	<40	26.78% (15)	73.22% (56)	1.56	0.66–3.67	0.388
	>40	14.67% (11)	85.33% (64)			
**Parameter**		**EGFR Positive**	**EGFR Negative**	**OR**	**CI**	** *p* ** **-Value**
Sex	Male	14.17% (17)	85.83% (103)	0.91	0.28–2.97	1.000
	Female	15.38% (4)	84.62% (22)			
Age	<40	15.49% (11)	84.51% (60)	1.19	0.47–3.00	0.815
	>40	13.33% (10)	86.67% (65)			

## Data Availability

The original contributions presented in this study are included in the article. Further inquiries can be directed to the corresponding author(s).

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
