# Peer review of "Cellular Rejection Post-Cardiac Transplantation: A 13-Year Single Unicentric Study"

_medicina, 2025, doi:10.3390/medicina61081317_

Round 1

Reviewer 1 Report

Comments and Suggestions for Authors

This is an interesting observational report on cellular rejection post cardiac transplantation . While the study provides valuable histopathologic data, its methodology has a number of problems.

Firstly, I see the paper lacks a clearly declared hypothesis and objective of the study. Secondly, no multivariate analysis for adjustment of confounding variables was performed.  I also found correlation claims are based solely on unadjusted p-values with no effect sizes or correlation coefficients being reported. The timing and frequency of biopsies are not reported, which would curtail clinical usefulness. Finally, biopsy results do not correlate with patient outcomes and hence the results are less relevant in practical life.

I have no doubts to elaborate on correlation of histopathologic observations with clinical outcomes would further enhance the translational applicability of observations. I would suggest the authors to mention these critical points . 

Reviewer 2 Report

Comments and Suggestions for Authors

1. The introduction should more clearly state the problem that the paper is intended to solve. And clearly state the purpose of the research. This is currently lacking.

2. The data selection process should be described in detail. How many total patients were operated on. How many were excluded and why? Why did the authors exclude patients who died? What is the time frame and cause of death for their exclusion? This requires clarification.

3. There is no ‘Statistical Analysis’ section. In Table 1 - what does the median age of 51-60 years mean? Is it the interquartile range? or what? and what was the median? You need to describe the data as is customary. Based on testing the hypothesis that the data are normally distributed.

4. The analyses presented in Table 2 are incorrect. Comparison of frequencies does not prove the relationship between the risk factor (Environment Rural) and the studied outcome (ACR). One-factor logistic regression analysis should be performed. And find the odds ratio.

5. Figure 1 has nothing to do with the presentation of results. I don't see the point of it.

6. Table 3. You also need to calculate the odds ratio. This applies to Table 4.1 and 4.2 as well.

7. Table 5. Why do the authors call this a correlation?  Correlation is the ratio of one quantitative parameter to another quantitative parameter.  In this case, the analysis includes binary values.

The authors did not describe the ‘statistical analysis’ section, which is a serious problem with the manuscript. The mathematical approach is not correct. The results obtained are questionable. The data must be recalculated using the correct analysis methods.

Authors should clearly understand what they want to achieve. At present, this appears to be simply descriptive statistics of a cohort of patients who have undergone heart transplantation. No association between any risk factors and treatment outcomes has been established. 
Phrase in the discussion: ‘These results may help optimise treatment strategies and improve outcomes for patients, providing a solid foundation for future research and the implementation of more effective clinical practices.’ 
What is the purpose of this statement? What outcomes are they optimising?

Round 2

Reviewer 1 Report

Comments and Suggestions for Authors

Thank you for addressing the previous comments and revising the manuscript accordingly. The authors have made substantial improvements in both clarity and content. Most of the prior concerns have been adequately addressed, and the revised version is more cohesive and well-structured. Minor language or formatting edits may be considered at the copyediting stage.

Reviewer 2 Report

Comments and Suggestions for Authors

I would recommend that the authors additionally contact a specialist in statistical analysis. Still, the approach to calculations is incomplete. Any quantitative indicator must be presented as an average (or median) value so are the variances (i.e., the spread) of the data. The authors demonstrate in this version of the article only the average age (56 years). And what is the variance? The standard deviation? Again, the authors calculated the odds ratio (OR), but do not show a 95% confidence interval. These points are essential for mathematical data analysis.
